# Innovation network structure, government R&D investment and regional innovation efficiency: Evidence from China

Xiao-Yan Cao[1,2], Xiang-Li Wu[1,2]*, Li-Min Wang[1,2]

1 College of Geographical Sciences, Harbin Normal University, Harbin, China, 2 Key Laboratory of Remote Sensing Monitoring of Geographic Environment of Heilongjiang Province, Harbin Normal University, Harbin, China

* jndxwxl@163.com

**Data Availability Statement:** All relevant data are within the paper and its Supporting Information files.

**Funding:** This work was supported by the Key Plan of National Social Science Foundation of China

## Abstract

Based on the panel data of 30 provinces in China from 2011 to 2019, this paper uses a two-stage DEA model to measure regional innovation efficiency, then non-parametric test is used to examine the impact of innovation network structure and government R&D investment on regional innovation efficiency. The results show that, at the provincial level, innovation efficiency of regional R&D is not necessarily in direct proportion to the innovation efficiency in the commercialization stage. Commercialization efficiency is not necessarily high in provinces with high technical R&D efficiency. At the national level, the innovation efficiency gap between our country's R&D and commercialization stage is small, indicating that the development of the national innovation efficiency is more and more balanced. Innovation network structure can promote the R&D efficiency, but has no significant effect on the commercialization efficiency. Government R&D investment helps to improve the R&D efficiency, but it is not conducive to the improvement of commercialization efficiency. The interaction between innovation network structure and government R&D investment will have compound effects on regional innovation efficiency; the region with underdeveloped innovation network structure can increase the government R&D investment to make it have a higher level of R&D. This paper provides insights into how to improve innovation efficiency in different social networks and policy environments.

## Introduction

Innovation has become an important means to improve the competitiveness of a country or region [1]. As the open innovation deepens, countries are actively integrating into the global innovation network, integrating science and technology resources, and increasing R&D spending. For example, in 2019, R&D spending in developed countries such as the United States, Germany and South Korea reached 3.2%, 3.2% and 4.6% of GDP, and the innovation index ranked 3rd, 9th and 11th respectively [2, 3]. In contrast, from 2010 to 2019, China's R&D investment increased from 1.7% to 2.2%, and the innovation index rose from 20th to

under the Grant 16BJY039. The funders had role in study design, data collection and analysis, decision to publish, or preparation of the manuscript.

**Competing interests:** The authors have declared that no competing interests exist.

14th [2, 3]. It can be seen that although China has made great progress in innovation input and output, but there is still a certain gap compared with the developed countries. Improving the efficiency of innovation and promoting the transformation of innovation in China, combining the need of national characteristics and global competition, is still a matter of great concern for researchers and policy makers.

Innovation efficiency is a key index to measure innovation achievements, which plays a very important role in decision-making [4]. Therefore, domestic and foreign scholars carry out research on the evaluation of regional innovation efficiency. However, many studies only focus on the efficiency evaluation from scientific and technological input to scientific and technological achievements output, and the evaluation on the transformation of scientific and technological achievements into economic benefits lacks depth [5]. Moreover, traditional research also ignores the fact that scientific research achievements may simultaneously play the roles of input and output in the innovation process, and at this time, the DMU will have an intermediate structure [6]. In addition, although the existing research has explored the influencing factors of regional innovation efficiency in terms of economic development level [7], foreign direct investment [8], research and development subsidy [9], fiscal decentralization [10], human capital [11], financial development [12], industrial structure [13] and many other aspects, it has neglected that all kinds of factors play a role in the context of the innovation environment. Generally, enterprises have higher innovation willingness in a good innovation environment, while a bad innovation environment may cause the "innovation paradox" of high innovation demand and low innovation capability [14].

Innovation efficiency is mainly influenced by social network environment and policy environment. The social network environment emphasizes the influence of the relationship between innovation subjects on innovation efficiency. In order to obtain the innovation resources beyond the boundary of their own organization, the innovators establish stable and sustainable formal or informal links to form the innovation network structure [15]. The results show that the efficiency of regional innovation is directly affected by the structure of regional innovation network. Innovation networks in regions with high innovation efficiency generally have the following characteristics: a large number of innovation network components, high degree of network openness, smooth connections, and strong sense of cooperation [16]. The policy environment emphasizes government support for innovation. The degree of government support is mainly measured by the intensity of government investment in R&D. Government R&D support is a policy tool to promote regional development, which is conducive to stimulating regional innovation potential [17]. The main means of government R&D support is the investment of R&D funds. Different types of government investment have different effects on innovation performance, but they all promote the improvement of regional innovation efficiency [18]. Data show that since the new century, China's R&D investment intensity continues to raise, from the scale of investment, China's total R&D funding now ranks second in the world.

It can be found that the existing studies have revealed the influence of innovation network structure and government R&D investment on innovation efficiency from different aspects, but there are still shortcomings: Firstly, the current researches on innovation network structure, government R&D investment and regional innovation efficiency mostly analyze innovation activities as a single process, and do not reveal the mechanism of each influencing factor on innovation in different development stages. Secondly, most studies only analyze the impact of innovation network structure or government R&D investment on innovation efficiency, and seldom pay attention to the compound effect of the two, which leads to the lack of deep understanding of innovation efficiency. In areas with underdeveloped innovation network structure, it is worth pondering whether the government can improve regional innovation

efficiency by increasing R&D investment. To this end, this paper uses the two-stage DEA model to measure innovation efficiency at the R&D and commercialization stages in 30 Chinese provinces. This study verifies the combined effects of innovation network structure and government R&D investment on regional innovation efficiency with the help of nonparametric tests. Finally, we empirically analyze the results of the study, hoping to provide theoretical reference and practical significance for China's innovation and development.

## Theoretical background and hypotheses

### Regional innovation efficiency

The term of innovation efficiency was first defined by United States scholar Joseph. Schumpeter was introduced by Chinese scholars and gradually attracted attention in China since the 1980s. At present, the research on regional innovation efficiency in China mainly focuses on the analysis of its spatio-temporal evolution characteristics and influencing factors. Domestic research is mainly carried out from the national [1], economic zone [19], urban agglomeration [20], provincial [21] and other levels. Chen Kaihua and Guan Jiancheng [22] believe that in the process of innovation transformation, only one fifth of regional innovation efficiency is good, and at the same time, most regional technological R&D capacity and commercialization capacity are significantly uncoordinated. Zhao Kaixu et al. [1] found that there is a gradient difference in innovation efficiency among the eastern, central and western regions of China, specifically "eastern> central> western," but the growth rate of central and western regions is greater than that of eastern regions. Sheng Yanwen et al. [7] took Yangtze River Delta, Pearl River Delta and Beijing-Tianjin-Hebei urban agglomerations as the research objects, and concluded that the innovation efficiency of Pearl River Delta is in the leading position, the innovation efficiency of Yangtze River Delta has the largest growth rate, and the innovation efficiency of Beijing-Tianjin-Hebei is relatively backward. Liu Hanchu et al. [23] compared and analyzed the innovation efficiency of various provinces in China, and pointed out that the innovation efficiency at the regional level shows a decreasing trend from east to west, and developed provinces such as Beijing, Shanghai, Guangdong and Tianjin belong to high-efficiency units, reflecting the coupling characteristics of economic development level and innovation efficiency to a certain extent. A large number of studies have confirmed the fact that there are differences in regional innovation efficiency in China. In terms of influencing factors, many scholars believe that regional economic development level, openness, government behavior, urbanization level, scientific and technological innovation resources and innovation environment have good explanatory power to the change and difference formation of innovation efficiency [1, 11, 14, 24].

The main methods of efficiency measurement are stochastic frontier analysis (SFA) and data envelopment analysis (DEA). Stochastic frontier analysis (SFA) is a parametric approach that decomposes changes in productivity into movements of the production possibility boundary and changes in technical efficiency, its most distinctive feature is that the error terms used to measure production differentials are decomposed into technical efficiency errors and random errors, which avoids the effect of statistical errors on efficiency measurements [25], but with SFA, only one index can be used to represent the innovation output, and innovation is a comprehensive process of multi-input and multi-output. It extends the concept of single-input and single-output engineering efficiency to the evaluation of the effectiveness of multi-input and multi-output decision making units (DMU), and data envelopment analysis the process of innovation, and in the reduction error, simplifies the algorithm simultaneously avoids the subjective factor influence [26]. Traditional DEA models treat innovation as a big system with only input and output and treat it as a "Black box" when evaluating innovation efficiency [19].

This approach ignores the internal structure of the innovation system and the internal operating mechanism, reducing the accuracy of the evaluation of innovation efficiency [27]. The process of innovation first manifests itself in technological innovation, and then in the transformation of scientific and technological achievements into economic returns. Therefore, we choose a two-stage DEA model to evaluate the innovation activities in two stages: technological R&D and commercialization, which is helpful to explain the innovation process of regional innovation system.

## Innovation network structure and regional innovation efficiency

Innovation network is a basic institutional arrangement to adapt to systematic innovation [28]. In the context of the basic system, the geographical division of labor and the linkage of production enterprises, research institutes and higher education institutions constitute a regional organizational system that supports and generates innovation [27]. The key to innovation networks is to stimulate innovation capacity and gain competitive advantage, emphasizing cooperation and complementarity among multi-innovation actors while possessing the structural and resource characteristics of complex networks themselves [29]. Network features, such as network size, density, openness, and network structure holes, have become key factors in the technological capability growth and innovation efficiency improvement of enterprises [16, 30]. Therefore, we believe that innovation network structure should be included in the study of innovation efficiency growth.

First, the innovation network structure has the characteristics of proximity [30]. On the one hand, the geographical proximity of innovation network structure promotes the collective action of enterprises, which in turn accelerates the knowledge flow between enterprises [31]. On the other hand, frequent face-to-face communication between enterprises close to each other deepens mutual trust, realizes resource sharing and improves innovation efficiency [32]. Geographic proximity also has a counter-effect, with excessive geographic proximity leading to spatial lock-in [33]. However, with the development of transportation and network, social proximity makes innovation gradually get rid of geographical constraints, and cross-regional cooperation among innovation subjects becomes possible. The social proximity of innovation network structure is increasingly considered as a key factor to promote knowledge flow and improve network performance [34]. Therefore, we believe that regions with developed innovation network structure have high R&D efficiency.

Second, the innovation network has realized the link to the innovation factor, reduced the innovation cost and accelerated the knowledge diffusion [15]. The "Optimality" of R&D personnel and the "Profit-seeking" of R&D Capital lead to the concentration of innovation elements in regions with more perfect innovation network structure, so as to form scale economy, and then promote regional innovation level [35]. Clusters with dense network structures have strong information transmission capabilities, and innovation agents can use innovation networks to obtain market information more easily and produce more market-oriented innovation outcomes [36]. The higher the density of innovation network is, the stronger the connection strength between network nodes is, which indicates that the motivation of collaborative innovation among innovation subjects is stronger, complementary information, knowledge resources, and so on, may be shared to a greater extent among heterogeneous innovators, contributing to the enhancement of innovation capacity and thus promoting the efficiency of regional knowledge transformation [27]. Based on the above analysis, we believe that regions with developed innovation network structure have high commercialization efficiency.

Taken together, the innovation network structure can have an important impact on regional innovation efficiency. The closer the constituent subjects of innovation network

structure are to each other, the more conducive to the exchange of information and mutual learning of knowledge, and the output of more R&D results. The clustering of innovation factors can reduce innovation costs and thus increase the economic value of knowledge achievements. In general, proximity and low interaction costs increase the likelihood of achieving better innovation efficiency under the same conditions. The assumptions of this paper are therefore as follows:

**Hypothesis 1.** Innovation network structure affects regional innovation efficiency.

**Hypothesis 1a.** Innovation network structure has a positive promoting effect on R&D efficiency.

**Hypothesis 1b.** Innovation network structure has a positive promoting effect on commercialization efficiency.

## Government R&D investment and regional innovation efficiency

Government R&D investment is the direct expenditure for the specific target selected by the government [37]. The investment object of these funds can be divided into universities, research institutions and enterprises according to the innovation subject [38]. Different types of government investment have different impacts on innovation performance, but they all have an impact on regional innovation efficiency [18]. There is still no consensus on the attitude of government R&D investment in academia. Some scholars believe that government R&D investment will have a positive incentive effect on regional innovation efficiency [39]. However, some scholars also found that the government's investment in innovation activities contributed to the inertia of enterprises engaged in innovation activities, resulting in crowding out effect [40]. The reason for the above two opposite conclusions may be that the innovation process is oversimplified, and the process of innovation output can be divided into two stages: technological output and economic output, and the influence of government R&D investment on innovation efficiency in the two stages may be different.

The first is that the technology spillover of R&D activities makes it impossible for enterprises to take all the returns of innovation factors, so the risk of market failure often occurs in the process of innovation investment [41]. As a special resource, government R&D investment provides financial support for enterprises' R&D activities, which effectively reduces the R&D cost and risk of enterprises in technological innovation [42]. When selecting R&D investment projects, the government will organize experts and professional evaluation bodies to select projects according to relevant regulations and regulations, thus ensuring the fairness of the evaluation process and the scientific nature of funded projects [43]. This makes government R&D investment can also be regarded as a credit endorsement, and enterprises can send positive financial signals to outside investors by obtaining government R&D investment with zero interest cost, thus enhancing their investment confidence [42]. Therefore, we believe that regions with high intensity of government R&D investment have high R&D efficiency.

Second, according to Samuelson's classical theory of public goods, it is inevitable that there will be inefficiency in government funding R&D activities with certain "public goods" attribute [40]. Moreover, when governments subsidize innovation, they tend to focus on the social benefits of innovation rather than on the direct economic benefits [37]. Government R&D investment is mostly applied to projects with long-term strategic significance, but this often makes it difficult to convert scientific research results into economic benefits [44]. At the same time, the attention of government leaders mainly focuses on the selection of

projects and neglects the later supervision, so the efficiency of government R&D investment cannot be guaranteed [45]. When enterprises rely too much on government support, or innovation is mostly public institutions, market regulation is difficult to play a role. Based on the above analysis, we believe that regions with high intensity of government R&D investment have low commercialization efficiency.

Taken together, increasing the intensity of government R&D investment can reduce firms' R&D risks, improve R&D confidence, and increase R&D outcomes. However, the directionality of government R&D investment can weaken the role of the market in innovation and limit business innovation. The assumptions of this paper are therefore as follows:

**Hypothesis 2.** Government R&D investment affects regional innovation efficiency.

**Hypothesis 2a.** Government R&D investment has a positive promoting effect on R&D efficiency.

**Hypothesis 2b.** Government R&D investment has a negative inhibiting effect on commercialization efficiency.

## Innovation network structure, government R&D investment and regional innovation efficiency

This paper aims to explore the impact of different social network and policy environments on the R&D efficiency and commercialization. Therefore, this paper proposes two hypotheses: different innovation network structure and government R&D investment background of regional innovation efficiency differences. However, due to China's special political and economic background, the development of enterprises, the construction of the network inevitably needs policy preferences. An isolated discussion of the structure of innovation networks and the role of government R&D investment cannot properly reveal complex social phenomena [46]. The node of innovation network and the object of government R&D investment are both the main body of innovation, so it is necessary to consider the comprehensive function of both. The effectiveness of R&D investment depends on the interaction between local knowledge producers and knowledge users, and the more frequent the interaction, the stronger the impact on innovation [47]. Generally speaking, the innovation network structure of low-level economic development regions is relatively backward. However, some studies have found that innovation efficiency is relatively higher in regions with small innovation networks but high government R&D Investment [48]. Based on the above point of view, this paper considers whether innovation can be attracted by increasing government investment, and whether small innovation networks can break out into high R&D efficiency when regional R&D investment is enough to catch up with big cities. The assumptions of this paper are therefore as follows:

**Hypothesis 3.** Innovation network structure and government R&D investment have a compound effect on regional innovation efficiency.

**Hypothesis 3a.** The R&D efficiency is relatively high in regions with underdeveloped innovation network structure but high government R&D investment intensity.

## Methodology

### Data sources

This paper selects panel data from 30 provinces of our country (except Hong Kong, Macao, Taiwan and Tibet) from 2011 to 2019 to analyze the impact of innovation network structure and government R&D investment on regional innovation efficiency. The data mainly come from the Statistical Yearbook of Science and Technology of China, the Statistical Yearbook of China, the Evaluation Report of Regional Innovation Ability of China, and the National Economic and Social Development Statistics Bulletin of the People's Republic of China. Considering that there is a time lag between the input and output of innovation efficiency, this paper, after referring to relevant studies [49, 50], concludes that the time lag between the input and the output of the first stage, a two-year time lag between phase I and phase II outputs is appropriate. Therefore, this paper constructs the data of five time points (2011-2013-2015, 2012-2014-2016, 2013-2015-2017, 2014-2016-2018, 2015-2017-2019), with a total of 150 decision making units.

### Measures

**Measurement of innovation efficiency.**  There is no unified understanding on the selection of innovation efficiency index variables so far. Drawing on relevant literature research [14, 19, 27], this paper constructs an index system for innovation efficiency evaluation. The internal expenditure of R&D funds is selected to reflect the actual regional investment in R&D funds, and the full-time equivalent of R&D personnel is used to represent the investment in personnel of regional innovation efficiency. Intermediates are measured by invention patents and scientific papers. Per capita GDP and per capita disposable income can reflect the improvement of people's lives and the promotion of regional economic level by the achievements of scientific research. Sales revenue of new products can reflect the direct economic benefits brought by the progress of science and technology to the relevant units. The transaction contract amount of technology market reflects the level of technology achievements into market value. So we chose these four as the indicators of output in the commercialization stage. The specific model for the two-stage efficiency evaluation is shown in Fig 1.

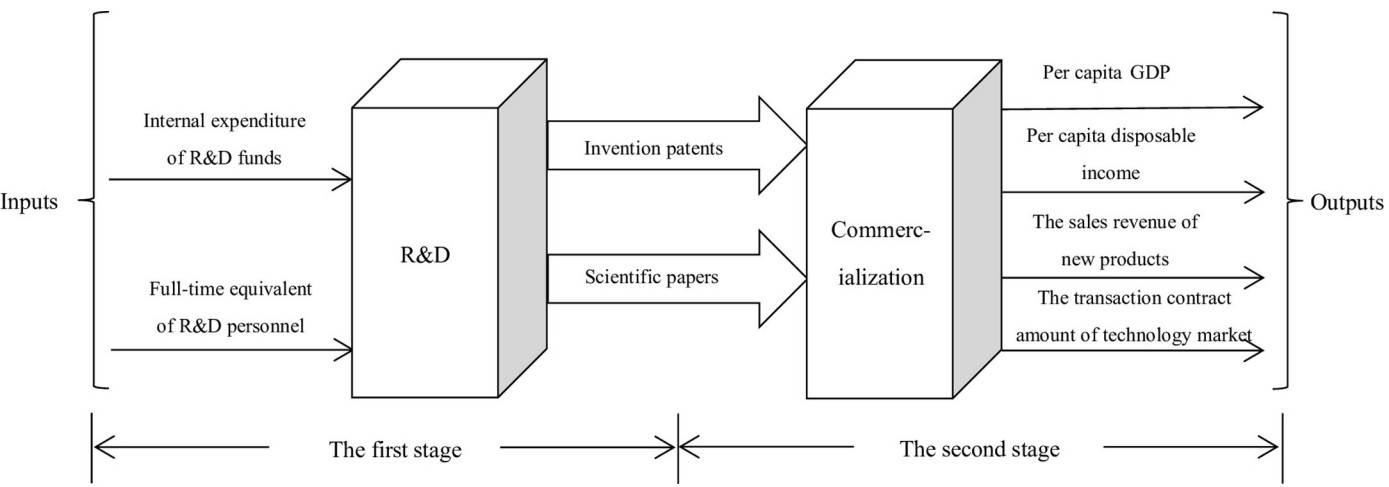

**Fig 1. Two-stage model of regional innovation efficiency evaluation.**

Considering the complexity of innovation process, this paper uses a two-stage DEA model to measure innovation efficiency [14]. There are n decision-making units, and each decision-making unit has m kinds of input vector X and s kinds of output vector Y. For any decision-making unit, the model representation is shown in the following form (1):

$$
\begin{cases}
\min\left(\theta - \varepsilon\left(\sum_{k=1}^{K}S^- + \sum_{l=1}^{L}S^+\right)\right) \\
s.t. \sum_{j=1}^{n}\lambda_{jk}X_j + S^- = \theta X_k^j (k = 1, 2, \cdots, K) \\
\sum_{j=1}^{n}\lambda_{jl}Y_j - S^+ = Y_l^j (l = 1, 2, \cdots, L) \\
\sum_{j=1}^{n}\lambda_j = 1 \\
\lambda_j \geq 0, j = 1, 2, \cdots, n \\
S^+ \geq 0, S^- \geq 0
\end{cases}
\tag{1}
$$

In this model, $S^+$, $S^-$ represents input redundancy or output insufficiency of decision making unit. $\varepsilon$ means infinitesimal. $\theta$ is the efficiency evaluation value of the DMU. when $\theta = 1$ and $S^{+} = S^- = 0$, DEA is effective; if $\theta = 1$ and $S^+ \neq 0$ or $S^- \neq 0$, the DEA is weakly effective; if $\theta < 0$, the DEA is invalid.

**Measurement of innovation network structure.** Based on the previous studies on the structure of regional innovation network [15, 51, 52], this paper evaluates the structure of innovation network from four dimensions: network scale, network openness, network structure hole and network link. The network scale is reflected by three indexes: the number of universities, the number of research and development institutions and the number of industrial enterprises above designated scale. The network openness is represented by the value of foreign technology import contracts and the number of foreign technology import contracts in each region. The self-raised funds of enterprises are used for the intramural expenditure on R&D in universities, the self-raised funds of enterprises are used for the intramural expenditure on R&D in research and development institutions, and the government funds are used for the intramural expenditure on R&D in industrial enterprises above designated size, these three indicators are selected to reflect the network links. The network structure is characterized by the number of public libraries, the number of employment agencies and the number of contracts in the technology market.

According to the above four aspects, according to the principle of data availability and representativeness, the evaluation index system of innovation network structure is constructed (see Table 1). The maximum and minimum value method is used to deal with the data dimensionless, and the entropy weight method is used to calculate the index weights of all levels. Finally, the comprehensive value of the innovation network structure of each region is determined. The above mathematical formula of entropy weight method can be found in Liang Lina and Yu Bo [15]. The length of the article is limited and will not be repeated here.

**Measurement of government R&D investment.** This paper introduces the government R&D investment intensity index [48]. The DMU is classified as government-oriented or non-government-oriented. The government R&D investment intensity index is calculated by dividing the proportion of government R&D spending in a region by the proportion of government R&D spending in the country. By combining government R&D expenditure data with national

R&D expenditure data, we can consider the relative proportion of R&D by a regional government, which can more comprehensively express the importance that the regional government attaches to innovation. The formula is as follows:

$$O = \frac{RDE_{kg}}{RDE_k} \bigg/ \frac{RDE_g}{RDE} \tag{2}$$

O is the government R&D investment intensity index, RDE is R&D expenditure in the nation, $RDE_K$ represents R&D expenditure in region, RDEg represents R&D expenditure of government, $RDE_{kg}$ represents R&D expenditure of government in the region.

**Non-parametric test.** Generally, when the data conform to normal distribution and the variance is homogeneous (or symmetrical), we usually use T test or F test. The nonparametric test is relative to the parametric test, which has no special requirements on the distribution of data. The nonparametric test has the advantages of simple and easy to master and robust conclusion [53]. When the parameter test is not applicable, the non-parameter method can use the data more effectively. Since the data in this paper don't conform to normal distribution, this study chooses to use non-parametric test.

The steps of hypothesis testing are as follows: ① Kolmogorov-Smirnov test is performed on the above variables to test the normality of the data and determine the correctness of using nonparametric test. ② Because Mann-Whitney U test is often used to test whether there is a significant difference between two independent unpaired samples, we use Mann-Whitney U test to test hypothesis 1 and hypothesis 2. ③ Kruskal-Wallis H test is used when there are two or more samples, which is an extension of Mann-Whitney U test, so we use Kruskal-Wallis H test to verify hypothesis 3.The above nonparametric test statistics and their mathematical formulas are detailed in Conover [54], which will not be repeated here due to the limited length of the article.

## Empirical analysis

### Measurement of regional innovation efficiency

In the research of regional innovation efficiency, we use DEAP2.1 software to calculate the innovation data from 2011 to 2019. Before proceeding with the analysis, we examine the statistics of the variables of innovation efficiency, and the results show that the standard deviation of most of the variables is greater than their average. As shown in Table 2, there are wide regional gaps in both inputs and outputs.

Table 3 shows the efficiency of innovation at the stage of technological R&D and commercialization by province. In the R&D stage, it can be seen that from 2013 to 2017, the R&D efficiency values of Beijing, Shaanxi and Gansu were all 1, and they showed a stable trend over a five-year period. The results show that the R&D efficiency of the three regions reaches DEA efficiency and the allocation of innovation resources reaches the optimal state. The R&D efficiency of Jilin, Heilongjiang, Chongqing, Sichuan and Guizhou is close to 1, indicating that the investment in capital and manpower has been fully utilized. The R&D efficiency of Tianjin, Liaoning, Shanghai, Jiangsu and Zhejiang provinces ranges from 0.6 to 0.9, indicating that the R&D efficiency of these regions is at a good level. The technological R&D efficiency of Hebei, Shanxi and other provinces is below 0.5, especially in Inner Mongolia, where the average R&D efficiency is 0.224. These regions need to improve the efficiency of knowledge innovation by improving the efficiency of innovation resources and avoiding unnecessary waste.

In the commercialization stage, it can be seen that Jilin, Guangdong, Qinghai and Ningxia have five years of commercialization efficiency of 1, reaching DEA effective, Inner Mongolia and Hunan have four years of commercialization efficiency of 1, and Beijing, Hebei, Jiangxi, Hubei and Hainan also have commercialization efficiency average close to 1. This indicates

**Table 1. Index system of innovation network structure.**

| Level 1 indicators | Secondary indicators | Weight | Three levels of indicators | Weight | Unit |
|---|---|---|---|---|---|
| Innovative network structure | Network size | 0.182 | Number of institutions of universities | 0.017 | Unit |
| | | | Number of research and development institutions | 0.024 | Unit |
| | | | Number of industrial enterprises above designated size | 0.141 | Unit |
| | Network openness | 0.296 | Value of foreign technology import contracts | 0.138 | 10,000 yuan |
| | | | Number of foreign technology import contracts | 0.158 | item |
| | Network link | 0.277 | The self-raised funds of enterprises are used for the intramural expenditure on R&D in universities | 0.083 | 10,000 yuan |
| | | | The self-raised funds of enterprises are used for the intramural expenditure on R&D in research and development institutions | 0.103 | 10,000 yuan |
| | | | The government funds are used for the intramural expenditure on R&D in industrial enterprises above designated size | 0.091 | 10,000 yuan |
| | Network structure hole | 0.245 | Number of public libraries | 0.058 | Unit |
| | | | Number of employment agencies | 0.129 | Unit |
| | | | Number of contracts in technology market | 0.058 | Unit |

that the conversion rate from science and technology inputs to economic output is relatively high in these regions and that innovation resources are being fully utilized. Compared with the R&D efficiency in the first stage, the innovation efficiency in Hebei, Inner Mongolia, Qinghai and Ningxia has improved greatly in the second stage. This indicates that although the efficiency of technological output in these regions is low, the efficiency of transforming research results into economic benefits is high. Meanwhile, compared with the higher R&D efficiency in the first stage, the efficiency values of Heilongjiang, Sichuan, Shaanxi and Gansu provinces in the second stage are significantly lower, indicating that the transformation rate of scientific and technological achievements is low, the original input of a large amount of innovation resources is not effectively transformed into economic results, and the contribution of scientific and technological innovation to regional economic growth is still far from adequate. The commercialization efficiency of Yunnan and Xinjiang is lower than 0.5. How to improve the overall level of regional innovation efficiency, which in turn drives economic development, is a pressing problem for these regions to solve.

In general, technological R&D does not necessarily correlate with innovation efficiency in the commercialization stage. Provinces with high technological R&D efficiency, such as Heilongjiang and Gansu, do not have high commercialization efficiency. This provides us with a new way to solve the problem of uneven distribution of regional innovation resources and

**Table 2. Descriptive statistics of innovation efficiency measurement indicators.**

| n = 150 | | Max | Min | Mean | SD |
|---|---|---|---|---|---|
| Inputs | Internal expenditure of R&D funds (10000 yuan) | 18012271 | 103717 | 3867121 | 4229844 |
| | Full-time equivalent of R&D personnel (10000 man-year) | 520303 | 4008 | 114174 | 120991 |
| Intermediates | Invention patents (piece) | 627834 | 1099 | 92032 | 119533 |
| | Scientific papers (piece) | 102763 | 214 | 16853 | 18765 |
| Outputs | Per capita GDP (yuan) | 164220 | 26165 | 61742 | 28013 |
| | Per capita disposable income(yuan) | 69441.6 | 13466.6 | 26437 | 10966 |
| | The sales revenue of new products (10000 yuan) | 42970.06 | 22.82 | 6174 | 8307 |
| | The transaction contract amount of technology market (10000 yuan) | 56952843 | 21861 | 4807018 | 8622204 |

**Table 3. Innovation efficiency of regional R&D in China.**

| Provinces | R&D efficiency | | | | | | Commercialization efficiency | | | | | |
|---|---|---|---|---|---|---|---|---|---|---|---|---|
| | 2013 | 2014 | 2015 | 2016 | 2017 | Mean | 2015 | 2016 | 2017 | 2018 | 2019 | Mean |
| Beijing | 1.00 | 1.00 | 1.00 | 1.00 | 1.00 | 1.000 | 0.74 | 0.81 | 1.00 | 1.00 | 1.00 | 0.909 |
| Tianjin | 0.73 | 0.64 | 0.60 | 0.75 | 0.54 | 0.652 | 0.88 | 0.79 | 0.53 | 0.50 | 0.85 | 0.707 |
| Hebei | 0.39 | 0.42 | 0.37 | 0.46 | 0.42 | 0.412 | 0.96 | 0.87 | 0.89 | 1.00 | 1.00 | 0.942 |
| Shanxi | 0.45 | 0.40 | 0.38 | 0.46 | 0.52 | 0.440 | 0.39 | 0.51 | 0.74 | 0.97 | 0.88 | 0.699 |
| Inner Mongolia | 0.22 | 0.22 | 0.20 | 0.25 | 0.23 | 0.224 | 0.93 | 1.00 | 1.00 | 1.00 | 1.00 | 0.985 |
| Liaoning | 0.72 | 0.65 | 0.60 | 0.64 | 0.70 | 0.662 | 0.51 | 0.60 | 0.64 | 0.89 | 0.72 | 0.673 |
| Jilin | 1.00 | 1.00 | 1.00 | 1.00 | 0.92 | 0.983 | 1.00 | 1.00 | 1.00 | 1.00 | 1.00 | 1.000 |
| Heilongjiang | 0.96 | 0.93 | 0.94 | 1.00 | 1.00 | 0.966 | 0.17 | 0.16 | 0.20 | 0.25 | 0.35 | 0.224 |
| Shanghai | 0.76 | 0.75 | 0.73 | 0.79 | 0.73 | 0.754 | 0.62 | 0.74 | 0.72 | 0.87 | 0.75 | 0.740 |
| Jiangsu | 1.00 | 0.88 | 0.61 | 0.77 | 0.69 | 0.789 | 0.56 | 0.62 | 0.61 | 0.64 | 0.59 | 0.604 |
| Zhejiang | 1.00 | 0.78 | 0.80 | 0.82 | 0.69 | 0.819 | 0.81 | 0.82 | 0.69 | 0.75 | 0.87 | 0.788 |
| Anhui | 1.00 | 0.85 | 0.77 | 0.96 | 0.88 | 0.892 | 0.65 | 0.66 | 0.66 | 0.67 | 0.65 | 0.657 |
| Fujian | 0.57 | 0.52 | 0.56 | 0.72 | 0.68 | 0.612 | 0.64 | 0.61 | 0.50 | 0.51 | 0.53 | 0.558 |
| Jiangxi | 0.51 | 0.67 | 0.62 | 1.00 | 1.00 | 0.759 | 0.93 | 0.94 | 0.90 | 0.88 | 1.00 | 0.929 |
| Shandong | 0.50 | 0.52 | 0.44 | 0.56 | 0.47 | 0.499 | 0.96 | 0.92 | 0.88 | 0.82 | 0.76 | 0.866 |
| Henan | 0.52 | 0.51 | 0.47 | 0.55 | 0.57 | 0.523 | 0.95 | 0.81 | 0.82 | 0.90 | 0.59 | 0.813 |
| Hubei | 0.64 | 0.69 | 0.70 | 0.75 | 0.71 | 0.697 | 0.92 | 0.89 | 0.89 | 1.00 | 0.95 | 0.932 |
| Hunan | 0.67 | 0.62 | 0.63 | 0.68 | 0.61 | 0.642 | 1.00 | 1.00 | 1.00 | 1.00 | 0.94 | 0.988 |
| Guangdong | 0.52 | 0.48 | 0.53 | 0.69 | 0.83 | 0.610 | 1.00 | 1.00 | 1.00 | 1.00 | 1.00 | 1.000 |
| Guangxi | 0.66 | 0.71 | 0.87 | 1.00 | 1.00 | 0.848 | 0.71 | 0.63 | 0.54 | 0.40 | 0.44 | 0.543 |
| Hainan | 0.67 | 0.58 | 0.68 | 0.73 | 0.80 | 0.691 | 0.81 | 0.91 | 1.00 | 1.00 | 0.82 | 0.906 |
| Chongqing | 1.00 | 1.00 | 1.00 | 0.95 | 0.81 | 0.952 | 0.81 | 0.72 | 0.58 | 0.70 | 0.63 | 0.687 |
| Sichuan | 0.91 | 0.85 | 0.78 | 0.98 | 1.00 | 0.904 | 0.33 | 0.29 | 0.33 | 0.41 | 0.70 | 0.411 |
| Guizhou | 0.98 | 1.00 | 0.83 | 0.91 | 1.00 | 0.942 | 0.40 | 0.51 | 0.44 | 0.61 | 1.00 | 0.592 |
| Yunnan | 0.72 | 0.72 | 0.67 | 0.79 | 0.66 | 0.711 | 0.33 | 0.34 | 0.40 | 0.44 | 0.39 | 0.380 |
| Shaanxi | 1.00 | 1.00 | 1.00 | 1.00 | 1.00 | 1.000 | 0.36 | 0.43 | 0.47 | 0.69 | 0.71 | 0.533 |
| Gansu | 1.00 | 1.00 | 1.00 | 1.00 | 1.00 | 1.000 | 0.53 | 0.40 | 0.43 | 0.37 | 0.51 | 0.446 |
| Qinghai | 0.24 | 0.31 | 0.40 | 0.57 | 0.68 | 0.438 | 1.00 | 1.00 | 1.00 | 1.00 | 1.00 | 1.000 |
| Ningxia | 0.47 | 0.41 | 0.45 | 0.52 | 0.62 | 0.496 | 1.00 | 1.00 | 1.00 | 1.00 | 1.00 | 1.000 |
| Xinjiang | 0.65 | 0.70 | 0.67 | 0.80 | 0.72 | 0.708 | 0.53 | 0.47 | 0.35 | 0.40 | 0.47 | 0.444 |
| Mean | 0.72 | 0.69 | 0.68 | 0.77 | 0.75 | 0.721 | 0.71 | 0.71 | 0.71 | 0.76 | 0.77 | 0.732 |

large gap in innovation efficiency. We can allocate innovation resources across regions from the national level to form the complementary advantages of scientific research highland and market-wide regions. On the whole, the average R&D efficiency of our country is 0.721, and the commercialization efficiency is 0.732. The difference between the two is small, which indicates that the development of innovation efficiency is more and more balanced in our country.

## Test the hypothesis

Before testing the hypothesis, we first test the normality of the data. The results show that: all indicators $P_{Kolmogorov-Smirnov} < 0.001$, that is, Innovation Network structure and Government R&D investment do not conform to the normal distribution, so the nonparametric test is used in this paper.

Then, we test whether the difference between R&D efficiency and commercialization efficiency is significant by Wilcoxon signed rank test. The Wilcoxon signed rank test is a test for

continuous variables with non-normal distribution. The test results are shown in Table 4. The results of Wilcoxon signed rank test show that there is no significant difference between R&D efficiency and commercialization efficiency. That is to say, the innovation efficiency of our country has basically reached equilibrium between the two stages.

Finally, we test the hypothesis and analyze the impact of innovation network structure and government R&D investment on regional innovation efficiency.

We conducted Mann-Whitney U test for hypothesis 1 and the results are shown in Table 5. At the R&D stage, there is a significant difference in innovation efficiency between regions with different levels of development of innovation network structure ($P_{\text{Mann-Whitney U}} = 0.011 < 0.05$). DMUs with well-developed innovation network structures have a high ranking and higher innovation efficiency values. This indicates that innovation network structure has a positive promoting effect on R&D efficiency, and hypothesis H1a was supported. This is mainly because R&D needs to be carried out in economically powerful large enterprises, research institutes and universities, which are the constituent subjects of the innovation network. Regions with well-developed innovation networks have stronger scale and connections of innovation subjects, which can better play the role of technological innovation. At the commercialization stage, the difference in innovation efficiency is not statistically significant ($P_{\text{Mann-Whitney U}} = 0.569 > 0.05$), and hypothesis H1b was rejected. This indicates that the innovation network structure does not influence the differences in innovation efficiency at the commercialization stage. The reason for this phenomenon may be that the state encourages mass entrepreneurship and innovation, and in regions with poor innovation network structures, the support of state policy funds is more likely to work to promote the development of SMEs. Although SMEs have limited capability in technological innovation, they also play a great role in transforming scientific research results into economic benefits, while we did not consider SMEs as the main body of the innovation network in our study due to their own limited innovation capability. In summary, it partially supports hypothesis H1 that there is a difference in innovation efficiency between regions with a developed innovation network structure and those with a less developed innovation network structure, and that the degree of development of the innovation network structure has a greater impact on regional R&D efficiency, while it has a smaller impact on commercialization efficiency.

We conducted the Mann-Whitney U test for hypothesis 2, and the results are shown in Table 6. There is a significant difference in innovation efficiency between regions with different intensity of government R&D investment in the R&D stage ($P_{\text{Mann-Whitney U}} = 0.000 < 0.01$). The efficiency ranking of DMUs with high government R&D investment intensity is higher than that of DMUs with low government R&D investment intensity, and the efficiency mean is higher than that of DMUs with low government R&D investment intensity. This indicates that government R&D investment has a significant positive effect on regional technology R&D, and hypothesis H2a was supported. Government R&D investment can make up for the lack of funds of innovation agents, reduce R&D risks, promote the improvement of regional innovation capacity, and have a positive driving effect on R&D innovation. In the commercialization stage, the results are reversed. DMUs with high intensity of government R&D investment rank low in efficiency and have low efficiency values ($P_{\text{Mann-Whitney U}} = 0.002 < 0.05$). This indicates that government R&D investment has a significant negative effect on commercialization and hypothesis H2b was supported. This implies that government intervention in R&D investment inhibits market dynamics and reduces the commercialization efficiency. In summary, hypothesis H2 was supported, and there is a difference in innovation efficiency between regions with high government R&D investment intensity and regions with low government R&D investment intensity, and regions with high government R&D investment intensity have high R&D efficiency but low commercialization efficiency. This finding is

**Table 4. Wilcoxon matched-pairs signed-rank test.**

| | n of DNU | Mean ranks | The sum of the rows |
|---|---|---|---|
| R&D stage > Commercialization stage | 69 | 68.70 | 4740.00 |
| R&D stage < Commercialization stage | 71 | 72.25 | 5130.00 |
| R&D stage = Commercialization stage | 10 | | |

also similar to the finding of Liu Xielin et al. [43], which once again proves the correctness of the test results.

Combining the two factors of innovation network structure and government R&D investment, we divide the research area into four groups. Group A indicates the area with developed innovation network structure and high government R&D investment, group B indicates regions with developed innovation networks but low government R&D investment, group C indicates regions with underdeveloped innovation networks and High Government R&D investment, and group D indicates regions with underdeveloped innovation networks, areas where the government spends less on research and development. We performed Kruskal-wallis H test for hypothesis 3 and post hoc test for each group, ranking the efficiency values of each group according to the modified significance level. The test results are shown in Table 7. There are differences in innovation efficiency between different groups ($P_{\text{Kruskal-Wallis H}} < 0.05$), hypothesis H3 is supported. The R&D efficiency is A > C > B > D. Group A has the highest R&D efficiency, which is the same as the hypothesis of H1a and H2a, that is, the region with advanced innovation network structure and high government R&D investment has the highest R&D efficiency. Government R&D investment can stimulate regional R&D innovation through innovation network from universities, R&D institutions, enterprises and other channels, and the two factors interact to promote R&D efficiency better. The R&D efficiency of group C was only lower than that of group A, and H3a was supported. This shows that in the regions with underdeveloped innovation network structures, increasing government R&D investment can stimulate local innovation vitality and keep the R&D efficiency at a high level. The R&D efficiency of group B and group D is poor, the structure of innovation network of these two groups is different, but the government R&D investment is lower, which shows that the government R&D investment has more influence in the R&D stage, and we should play the role of government better.

The commercialization efficiency is B > D > A > C. The commercialization efficiency in group B and group D is higher than that in group A and group C, which is the same as that in H2b. That is, the efficiency of regional commercialization is higher when the government R&D investment is low. The government's role in transforming the economy is limited, and the market plays a decisive role in that process. The commercialization efficiency of group B is the highest, while that of group C is the lowest, which shows that the innovation network structure is the carrier of the innovation process, innovation networks can enhance the impact of such factors on the innovation process.

**Table 5. Mann-Whitney U test of innovation network structure.**

| | Innovative Network structure | N of DMU | Mean ranks | Mean EFF value | Mann-Whitney U | Z | P |
|---|---|---|---|---|---|---|---|
| R&D stage | Developed | 59 | 60.48 | 0.779 | 2038.500 | -2.550 | 0.011 |
| | Underdeveloped | 91 | 79.82 | 0.682 | | | |
| Commercialization stage | Developed | 61 | 69.10 | 0.756 | 2552.500 | -0.570 | 0.569 |
| | Underdeveloped | 89 | 72.28 | 0.716 | | | |

**Table 6. Mann-Whitney U test of government R&D investment.**

| | Government investment in R&D | N of DMU | Mean ranks | Mean EFF value | Mann-Whitney U | Z | P |
|---|---|---|---|---|---|---|---|
| R&D stage | High | 59 | 45.93 | 0.834 | 1285.500 | -5.409 | 0.000 |
| | Low | 91 | 89.04 | 0.647 | | | |
| Commercialization stage | High | 59 | 84.71 | 0.640 | 1870.000 | -3.157 | 0. 002 |
| | Low | 91 | 62.12 | 0.792 | | | |

## Conclusions and revelation

### Conclusion

It is an important research content in the field of regional innovation to reveal the differences of regional innovation and find out the influencing factors of regional differences. Using two-stage DEA method, this paper measures the R&D stage and commercialization stage in 30 provinces of China, and analyzes the spatial and temporal differences of the R&D efficiency and commercialization efficiency. Based on the previous research, this paper tries to find out the key environmental factors that affect the efficiency of innovation. The influence of innovation network structure and government R&D investment on innovation efficiency is analyzed by non-parameter test. The main conclusions are as follows:

At the provincial level, the R&D efficiency is not necessarily in direct proportion to the commercialization efficiency. Commercialization efficiency is not necessarily high in provinces with high R&D efficiency. At the national level, the gap of innovation efficiency between our country's R&D and commercialization stage is small, which indicates that the development of innovation efficiency is more and more balanced.

The test of hypothesis 1 shows that the innovation network structure has a positive effect on the R&D efficiency, but has no significant effect on the commercialization efficiency. This shows that the establishment of close links between the main bodies of innovation networks, breaking the barriers of resource sharing, will help to improve the R&D efficiency. However, in the process of knowledge achievement transforming into economic output, the role of public innovators, such as universities and scientific research institutes, is negligible, and more depends on the popularization of knowledge in the society and the participation of enterprises.

**Table 7. Kruskal-Wallis H test.**

| | Innovative Network structure | Government investment R&D | Group code | N of DMU | Mean ranks | Mean EFF value | Kruskal-Wallis H | Post-hoc test (sig) |
|---|---|---|---|---|---|---|---|---|
| R&D stage | Developed | High | A | 21 | 29.86 | 0.904 | 38.525*** (df = 3) | A-B*** A-C** A-D*** B-C** B-D** C-D*** |
| | | Low | B | 40 | 76.68 | 0.713 | | |
| | Underdeveloped | High | C | 38 | 54.82 | 0.795 | | |
| | | Low | D | 51 | 98.75 | 0.596 | | |
| Commercialization stage | Developed | High | A | 21 | 88.14 | 0.652 | 10.524 ** (df = 3) | A-B** A-C A-D** B-C** B-D C-D** |
| | | Low | B | 40 | 58.33 | 0.815 | | |
| | Underdeveloped | High | C | 38 | 82.82 | 0.633 | | |
| | | Low | D | 51 | 65.10 | 0.773p | | |

Note: * p < 0.1,

** p < 0.05,

*** p < 0.01; post hoc test adjusted significance values by Bonferroni correction.

The test results of hypothesis 2 show that government R&D investment helps to improve the R&D efficiency, but it is not beneficial to the commercialization efficiency. This shows that the government still plays a very important role in the innovation process. The government sends a positive signal to the society by means of financial support, which increases the confidence of the region in technological research and development. In the period of commercialization, too much government intervention will restrain the vitality of the market. Therefore, the government should have a clear position for itself, support R&D activities with long-term social benefits, and leave more economic activities aimed at short-term benefits to the market.

The test results of hypothesis 3 show that regions with underdeveloped innovation network structures can achieve a higher level of R&D by increasing government R&D investment. This proves that the interaction between innovation network structure and government R&D investment has a compound effect on regional innovation efficiency. The government's financial support can attract more R&D organizations to the region, and can also stimulate the technological R&D potential of innovation network agents, making them the driving force of regional innovation development, but a web of innovators who rely too heavily on government direction can lose sight of what the market needs. Therefore, it is necessary to strike a proper balance between network construction and government R&D investment in order to ensure that knowledge results can be efficiently transformed into economic output.

## Revelation

The results of this paper provide some insights into how to improve innovation efficiency in different social network and policy environments.

In order to improve regional innovation efficiency, we need to build a sound innovation network structure. Expand the scale of the innovation network and form an innovation network mainly composed of enterprises, supplemented by universities, research institutions, governments, and technology intermediaries. We should make innovation networks more open, fully absorb foreign investment, enhance the sharing of innovation knowledge and achievements, and break down trade barriers. Strengthen the innovation network link and build the industry-university-research collaborative innovation mode, so as to improve the spontaneity, stability and sustainability of cooperation among innovation subjects. We should increase the number of structural holes in innovation networks, improve the construction of science and technology service platforms, promote the free flow of innovation information, and reduce the information asymmetry among innovation subjects.

In order to better leverage the government's role, we need to optimize the innovation policy environment and reasonably allocate government R&D investment. Actively play the government's functions in innovation strategy leadership, innovation environment construction and direct participation to lead the region to achieve innovation-driven development. Optimize the allocation of government R&D investment among innovation subjects. With the deepening of marketization, gradually cut the separate funding for research institutions and focus on supporting the industry-university-research collaborative innovation mode. We should explore a coordinated supervision and restraint mechanism for innovation investment subjects, so as to improve the transparency of the government's behavior in allocating innovation resources on the one hand and ensure the efficient use of government R&D investment on the other hand.

We divided the study areas into four groups: group A, where the innovation network structure is developed, and the government R&D investment is high, and group B, where the innovation network structure is developed but the government R&D investment is low, group C indicates regions with underdeveloped innovation networks and high government R&D

investment, while group D indicates regions with underdeveloped innovation networks and low government R&D investment. The results of nonparametric test show that R&D Efficiency A > C > B > D, commercialization efficiency B > D > A > C. Groups A and C have high R&D efficiencies that do not translate into high economic output, and these regions should be better served by markets and less government intervention. Perfect product market development, the original government control of the product can be determined by the market price to the market decision [55]. The investment strategy is appropriately tilted towards start-ups and small businesses in strategic industries. Group B is less efficient in technology R&D. Therefore, the focus should be on how to enhance the R&D efficiency, strengthen the linkages among innovation agents, reduce the cost of interaction and break down the barriers of communication We should improve the knowledge conversion rate of universities and scientific research institutes, in order to bring the knowledge spillover effect of universities and scientific research institutions into full play, each regional government should play the role of "Coordinator", guide all parties of Industry, education and research in the region to cooperate and innovate, and increase the proportion of R&D funds to regional GDP [27]. There is great potential for improving the efficiency of R&D in group D, which is the region most in need of increasing government investment in R&D. By sending positive signals, the government can attract research institutes and expand the scale of regional innovation networks. To sum up, the research results show that the combination of improving the innovation network structure and increasing government R&D investment can make the regional innovation efficiency increase in equilibrium. Regions with low R&D efficiency need more support from government R&D investment. Regions with low commercialization efficiency should pay attention to the regulating role of the market. In regions with underdeveloped innovation network structure, we should consider how to attract innovators by government R&D investment and establish efficient innovation network.

## Discussion

The contribution of this paper is: Firstly, we use the two-stage DEA to measure the efficiency of regional innovation, which has the advantage of dividing innovation activities into two stages: technological R&D and commercialization, the "Black Box" of the innovation system was broken by using the first stage of scientific and technological output as the input of the second stage. Secondly, we creatively analyze the innovation efficiency in the stage of technological R&D and commercialization from the perspective of innovation network structure and Government R&D investment, and pass the non-parametric test, to verify the difference of innovation efficiency in different stages of innovation activities between regions with different innovation network structure and government R&D investment, it is found that regions with backward innovation network structure can also stimulate the development of innovation by increasing government R&D investment. Thirdly, this paper proves that the combination of improving the innovation network structure and increasing the government R&D investment can make the regional innovation efficiency balanced, so as to provide guidance for the development of regional innovation differentiation in our country.

The conclusion of this paper not only provides some reference and guidance for theoretical research and practical application, but also has some limitations. First, we only study the impact of two key environmental factors, innovation network structure and government R&D investment, and do not fully consider the impact of other factors, such as industrial structure, economic development level, foreign investment and so on. Secondly, because of the difficulty of data acquisition, we take the province as the research unit, and cannot analyze the spatial

difference of the region in more detail. In the future research, we will carry out more full empirical analysis.

## Supporting information

**S1 Data.**
(XLSX)

## Author Contributions

**Conceptualization:** Xiao-Yan Cao.

**Data curation:** Xiao-Yan Cao.

**Formal analysis:** Xiao-Yan Cao.

**Funding acquisition:** Xiang-Li Wu.

**Investigation:** Xiao-Yan Cao.

**Methodology:** Xiao-Yan Cao.

**Project administration:** Xiao-Yan Cao.

**Resources:** Xiao-Yan Cao, Xiang-Li Wu, Li-Min Wang.

**Software:** Xiao-Yan Cao, Xiang-Li Wu, Li-Min Wang.

**Supervision:** Xiao-Yan Cao, Xiang-Li Wu.

**Validation:** Xiao-Yan Cao.

**Writing – original draft:** Xiao-Yan Cao.

**Writing – review & editing:** Xiao-Yan Cao.

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
