## [Decision Letter · Decision Letter 0]

18 Apr 2023

PONE-D-23-08235Innovation network structure, government R&D investment and regional innovation efficiency: Evidence from ChinaPLOS ONE

Dear Dr. Wu,

Thank you for submitting your manuscript to PLOS ONE. After careful consideration, we feel that it has merit but does not fully meet PLOS ONE’s publication criteria as it currently stands. Therefore, we invite you to submit a revised version of the manuscript that addresses the points raised during the review process.

We look forward to receiving your revised manuscript.

Kind regards,

Qingsong He, Ph.D.

Academic Editor

PLOS ONE

Journal Requirements:

Reviewers' comments:

Reviewer's Responses to Questions

**Comments to the Author**

1. Is the manuscript technically sound, and do the data support the conclusions?

Reviewer #1: Yes

Reviewer #2: Partly

2. Has the statistical analysis been performed appropriately and rigorously? 

Reviewer #1: Yes

Reviewer #2: Yes

3. Have the authors made all data underlying the findings in their manuscript fully available?

Reviewer #1: Yes

Reviewer #2: Yes

4. Is the manuscript presented in an intelligible fashion and written in standard English?

Reviewer #1: Yes

Reviewer #2: Yes

5. Review Comments to the Author

Reviewer #1: This paper studys the relationship among innovation network, government RD investment and regional innovation efficiency including commercialization, the results sound right as dividing samples into four groups of A,B,C and D, and some deeper reasons are in want to explain why.

Reviewer #2: Dear Author,

Thank you for submitting your manuscript entitled "Innovation network structure, government R&D investment and regional innovation efficiency: Evidence from China" to PLOS ONE. After reviewing the manuscript, I have the following comments and suggestions:

1.Introduction：

The introduction section provides a comprehensive overview of the connotation and influencing factors of innovation efficiency, pointing out the shortcomings of existing research and the contributions of this article. While the content is detailed, it also makes this section too lengthy. I recommend the author to add a DISCUSS chapter to expound the contributions and shortcomings of the article

2.Theoretical background and hypotheses：

In this section, the author elaborates on the article from four aspects: Regional innovation efficiency、Innovation network structure and regional innovation efficiency、Government R&D investment and regional innovation efficiency、Innovation network structure, government R&D investment and regional innovation efficiency,and propose reasonable assumptions separately. I think the hypotheses presented in this paper are well-formed and relevant to the research objectives. However, some of them need further elaboration to clarify their connections with the theoretical framework. Additionally, it would be helpful if you could clearly state the expected direction of each hypothesis (positive or negative) for easier interpretation.

3.Methodology：

I appreciate the detailed descriptions of each indicator measurement process. However, I believe that some of these details are excessive and could be simplified to enhance readability without sacrificing information. The lengthy explanation for each indicator may cause unnecessary confusion for readers who are not familiar with the specific measurement process. As a result, I suggest that you consider revising this section to summarize the key steps in a more concise and straightforward manner.

Overall, please revise the indicators measurement section to balance thoroughness and clarity so that readers can easily follow your methodology without being overwhelmed by unnecessary detail.

4.Empirical part：

I found that the paper lacks a dedicated section for empirical analysis. For instance, the measurement of regional innovation efficiency and hypothesis testing can be combined to form an empirical analysis section, which is missing in the current manuscript. This section is essential for substantiating the arguments made in the earlier sections of the paper.

Therefore, I recommend that you revise your manuscript to include a comprehensive empirical analysis section.

5.Conclusion：

In this section,the authors have effectively utilized their hypotheses to substantiate their arguments, providing valuable insights that can be useful for academics and policymakers interested in this field. But I suggest that the author can provide a more macro policy suggestion from an overall perspective to improve the summary.

6. PLOS authors have the option to publish the peer review history of their article (what does this mean?). If published, this will include your full peer review and any attached files.

Reviewer #1: No

Reviewer #2: No

---

## [Author Response · Author response to Decision Letter 0]

4 May 2023

Thank you for your letter and for the reviewers' and editors' comments concerning our manuscript. These comments are very helpful for revising and improving our paper. We have studied comments carefully and have made corrections which we hope meet with approval.

Reviewer #1 comments: 

This paper studys the relationship among innovation network, government RD investment and regional innovation efficiency including commercialization, the results sound right as dividing samples into four groups of A,B,C and D, and some deeper reasons are in want to explain why.

Response: 

Thank you for your comment of this paper. Based on the panel data of 30 provinces in China from 2011 to 2019, this paper uses a two-stage DEA model to measure regional innovation efficiency, and uses non-parametric empirical method to test the impact of innovation network structure and government R&D input on regional innovation efficiency. The innovation of this paper is that on the basis of the original research, the research area is divided into groups A,B,C and D, hoping to give more targeted suggestions for different types of areas.

Reviewer #2 comments1: 

1.Introduction：

The introduction section provides a comprehensive overview of the connotation and influencing factors of innovation efficiency, pointing out the shortcomings of existing research and the contributions of this article. While the content is detailed, it also makes this section too lengthy. I recommend the author to add a DISCUSS chapter to expound the contributions and shortcomings of the article:

Response:

Thanks for your valuable comments and suggestions. We have revised the introduction. First, we simplified the description of the research status of innovation efficiency. Second, this paper first introduces the research status of innovation efficiency, and points out the shortcomings of existing research. Then we hope to supplement the existing research through this paper. There is a logical sequence, therefore, we did not delete the shortcomings of the existing research. Third, we moved the contribution to the end of the article and included a discussion section. Fourth, we selectively deleted meaningless sentences and paragraphs in the article. 

Reviewer #2comments2:

2.Theoretical background and hypotheses：

In this section, the author elaborates on the article from four aspects: Regional innovation efficiency、Innovation network structure and regional innovation efficiency、Government R&D investment and regional innovation efficiency、Innovation network structure, government R&D investment and regional innovation efficiency,and propose reasonable assumptions separately. I think the hypotheses presented in this paper are well-formed and relevant to the research objectives. However, some of them need further elaboration to clarify their connections with the theoretical framework. Additionally, it would be helpful if you could clearly state the expected direction of each hypothesis (positive or negative) for easier interpretation.

Response:

Thanks for your valuable comments and suggestions. We think your comments are very constructive. While introducing the theoretical background, we focus on the theory itself but ignore to clarify their connections with the theoretical framework. The specific hypotheses don’t succinctly state whether the structure of innovation networks and government R&D investment have positive or negative effects on innovation efficiency. We have revised the theoretical background and hypotheses. First, we add a paragraph to the conclusion of the innovation network section and the government R&D investment section respectively. Second, we reformulate the specific assumptions. The details are as follows:

Innovation network structure and regional innovation efficiency

……

First, the innovation network structure has the characteristics of proximity [30]. On the one hand, the geographical proximity of innovation network structure promotes the collective action of enterprises, which in turn accelerates the knowledge flow between enterprises [31]. On the other hand, frequent face-to-face communication between enterprises close to each other deepens mutual trust, realizes resource sharing and improves innovation efficiency [32]. Geographic proximity also has a counter-effect, with excessive geographic proximity leading to spatial lock-in [33]. However, with the development of transportation and network, social proximity makes innovation gradually get rid of geographical constraints, and cross-regional cooperation among innovation subjects becomes possible. The social proximity of innovation network structure is increasingly considered as a key factor to promote knowledge flow and improve network performance [34]. Therefore, we believe that regions with developed innovation network structure have high R&D efficiency.

Second, the innovation network has realized the link to the innovation factor, reduced the innovation cost and accelerated the knowledge diffusion [15]. The "Optimality" of R&D personnel and the "Profit-seeking" of R&D Capital lead to the concentration of innovation elements in regions with more perfect innovation network structure, so as to form scale economy, and then promote regional innovation level [35]. Clusters with dense network structures have strong information transmission capabilities, and innovation agents can use innovation networks to obtain market information more easily and produce more market-oriented innovation outcomes [36]. The higher the density of innovation network is, the stronger the connection strength between network nodes is, which indicates that the motivation of collaborative innovation among innovation subjects is stronger, complementary information, knowledge resources, and so on, may be shared to a greater extent among heterogeneous innovators, contributing to the enhancement of innovation capacity and thus promoting the efficiency of regional knowledge transformation [27]. Based on the above analysis, we believe that regions with developed innovation network structure have high commercialization efficiency.

Taken together, the innovation network structure can have an important impact on regional innovation efficiency. The closer the constituent subjects of innovation network structure are to each other, the more conducive to the exchange of information and mutual learning of knowledge, and the output of more R&D results. The clustering of innovation factors can reduce innovation costs and thus increase the economic value of knowledge achievements. In general, proximity and low interaction costs increase the likelihood of achieving better innovation efficiency under the same conditions. The assumptions of this paper are therefore as follows:

Hypothesis 1. Innovation network structure affects regional innovation efficiency. 

Hypothesis 1a. Innovation network structure has a positive promoting effect on R&D efficiency. Hypothesis 1b. Innovation network structure has a positive promoting effect on commercialization efficiency. 

Government R&D investment and regional innovation efficiency

……

The first is that the technology spillover of R&D activities makes it impossible for enterprises to take all the returns of innovation factors, so the risk of market failure often occurs in the process of innovation investment [41]. As a special resource, government R&D investment provides financial support for enterprises' R&D activities, which effectively reduces the R&D cost and risk of enterprises in technological innovation [42]. When selecting R&D investment projects, the government will organize experts and professional evaluation bodies to select projects according to relevant regulations and regulations, thus ensuring the fairness of the evaluation process and the scientific nature of funded projects [43]. This makes government R&D investment can also be regarded as a credit endorsement, and enterprises can send positive financial signals to outside investors by obtaining government R&D investment with zero interest cost, thus enhancing their investment confidence [42]. Therefore, we believe that regions with high intensity of government R&D investment have high R&D efficiency.

Second, according to Samuelson's classical theory of public goods, it is inevitable that there will be inefficiency in government funding R&D activities with certain "public goods" attribute [40]. Moreover, when governments subsidize innovation, they tend to focus on the social benefits of innovation rather than on the direct economic benefits [37]. Government R&D investment is mostly applied to projects with long-term strategic significance, but this often makes it difficult to convert scientific research results into economic benefits [44]. At the same time, the attention of government leaders mainly focuses on the selection of projects and neglects the later supervision, so the efficiency of government R&D investment cannot be guaranteed [45]. When enterprises rely too much on government support, or innovation is mostly public institutions, market regulation is difficult to play a role. Based on the above analysis, we believe that regions with high intensity of government R&D investment have low commercialization efficiency. 

Taken together, increasing the intensity of government R&D investment can reduce firms' R&D risks, improve R&D confidence, and increase R&D outcomes. However, the directionality of government R&D investment can weaken the role of the market in innovation and limit business innovation. The assumptions of this paper are therefore as follows:

Hypothesis 2. Government R&D investment affects regional innovation efficiency.

Hypothesis 2a. Government R&D investment has a positive promoting effect on R&D efficiency.

Hypothesis 2b. Government R&D investment has a negative inhibiting effect on commercialization efficiency.

Innovation network structure, government R&D investment and regional innovation efficiency

……The assumptions of this paper are therefore as follows:

Hypothesis 3. Innovation network structure and government R&D investment have a compound effect on regional innovation efficiency. 

Hypothesis 3a. The R&D efficiency is relatively high in regions with underdeveloped innovation network structure but high government R&D investment intensity.

Thanks! 

Reviewer #2comments3:

3.Methodology：

I appreciate the detailed descriptions of each indicator measurement process. However, I believe that some of these details are excessive and could be simplified to enhance readability without sacrificing information. The lengthy explanation for each indicator may cause unnecessary confusion for readers who are not familiar with the specific measurement process. As a result, I suggest that you consider revising this section to summarize the key steps in a more concise and straightforward manner.

Overall, please revise the indicators measurement section to balance thoroughness and clarity so that readers can easily follow your methodology without being overwhelmed by unnecessary detail.

Response: 

Thanks for your valuable comments and suggestions. We have revised the methods section according to your suggestions. The specific changes and responses are as follows:

(i) On the measurement of innovation efficiency, we simplified the introduction of the index system and deleted some explanations of specific indicators. For the expressive aspect of DEA method, we simplified the interpretation of formulas so that readers can understand the article more easily. We hope to increase the readability of the article. 

(ii) The rest of the changes are minor, with only some specific statement expressions being optimized. In these parts, we have simplified the expression in the original text. For example, entropy weight method and nonparametric test method don’t list detailed formulas. The index system of government R&D investment is put forward in the research method. 

Reviewer #2comments4:

4.Empirical part：

I found that the paper lacks a dedicated section for empirical analysis. For instance, the measurement of regional innovation efficiency and hypothesis testing can be combined to form an empirical analysis section, which is missing in the current manuscript. This section is essential for substantiating the arguments made in the earlier sections of the paper.

Therefore, I recommend that you revise your manuscript to include a comprehensive empirical analysis section.

Response:

Thank you for your valuable comments and suggestions. We have reconsidered the article and found your comments very constructive. Therefore, according to your suggestions, we combine the measurement of regional innovation efficiency with hypothesis testing to form the empirical analysis part.

Reviewer #2comments5:

5.Conclusion：

In this section,the authors have effectively utilized their hypotheses to substantiate their arguments, providing valuable insights that can be useful for academics and policymakers interested in this field. But I suggest that the author can provide a more macro policy suggestion from an overall perspective to improve the summary.

Response:

Thank you for your valuable comments and suggestions. We add more macro policy recommendations based on the conclusions of this paper. The details are as follows:

Revelation

The results of this paper provide some insights into how to improve innovation efficiency in different social network and policy environments. 

In order to improve regional innovation efficiency, we need to build a sound innovation network structure. Expand the scale of the innovation network and form an innovation network mainly composed of enterprises, supplemented by universities, research institutions, governments, and technology intermediaries. We should make innovation networks more open, fully absorb foreign investment, enhance the sharing of innovation knowledge and achievements, and break down trade barriers. Strengthen the innovation network link and build the industry-university-research collaborative innovation mode, so as to improve the spontaneity, stability and sustainability of cooperation among innovation subjects. We should increase the number of structural holes in innovation networks, improve the construction of science and technology service platforms, promote the free flow of innovation information, and reduce the information asymmetry among innovation subjects.

In order to better leverage the government's role, we need to optimize the innovation policy environment and reasonably allocate government R&D investment. Actively play the government's functions in innovation strategy leadership, innovation environment construction and direct participation to lead the region to achieve innovation-driven development. Optimize the allocation of government R&D investment among innovation subjects. With the deepening of marketization, gradually cut the separate funding for research institutions and focus on supporting the industry-university-research collaborative innovation mode. We should explore a coordinated supervision and restraint mechanism for innovation investment subjects, so as to improve the transparency of the government's behavior in allocating innovation resources on the one hand and ensure the efficient use of government R&D investment on the other hand.

Thanks once more!

Editors #comments1: 

Response: 

Thank you very much for your reminder. We have rechecked and corrected the references in the manuscript.

---

## [Editor Report · Decision Letter 1]

9 May 2023

Innovation network structure, government R&D investment and regional innovation efficiency: Evidence from China

PONE-D-23-08235R1

Dear Dr. Wu,

We’re pleased to inform you that your manuscript has been judged scientifically suitable for publication and will be formally accepted for publication once it meets all outstanding technical requirements.

Kind regards,

Qingsong He, Ph.D.

Academic Editor

PLOS ONE
---

## [Editor Report · Acceptance letter]

11 May 2023

PONE-D-23-08235R1 

Innovation network structure, government R&D investment and regional innovation efficiency: Evidence from China 

Dear Dr. Wu:

I'm pleased to inform you that your manuscript has been deemed suitable for publication in PLOS ONE. Congratulations! Your manuscript is now with our production department. 

Kind regards, 

on behalf of

Dr. Qingsong He 

Academic Editor

PLOS ONE